# Influences of sudden stratospheric warmings on the ionosphere above Okinawa

Klemens Hocke[1,2], Wenyue Wang[1,2], and Guanyi Ma[3]

[1]Institute of Applied Physics, University of Bern, 3012 Bern, Switzerland
[2]Oeschger Centre for Climate Change Research, University of Bern, 3012 Bern, Switzerland
[3]National Astronomical Observatories, Chinese Academy of Sciences, Beijing 100101, China

**Correspondence:** Klemens Hocke (klemens.hocke@unibe.ch)

**Abstract.** We analyzed the ionosonde observations at Okinawa (26.7°N, 128.1°E, magnetic latitude: 17.0°N) for the years from 1972 to 2023. Okinawa is in the northern low latitude ionosphere where the influences of sudden stratospheric warmings (SSWs) on the ionosphere are expected to be stronger than in the mid- and high latitude ionosphere. We divided the dataset into winters with major SSWs in the northern hemisphere (SSW years) and into winters without major SSWs (no SSW years). During the SSW years, the daily cycle of the $F2$ region electron density maximum ($NmF2$) was stronger than in the no SSW years. The relative $NmF2$ amplitudes of solar and lunar tidal components ($S_2$, $O_1$, $M_2$, $MK_3$) are stronger by 3 to 8% compared to no SSW years. The semidiurnal amplitude, averaged for 29 SSW events, has a significant peak at the central date of the SSW (epoch time 0 of the composite analysis). The SSW influence is not strong: the semidiurnal amplitude is about 38.2% in SSW years and about 34.0% in no SSW years (relative to $NmF2$ of the background ionosphere). However, there is a sharp decrease of the amplitude by about 10% after the SSW peak was reached. The amplitude of the diurnal component does not show a single peak at the central date of the SSW. We present the maximal semidiurnal amplitudes of the SSWs since 1972. The SSW of 31 December 1984 has the strongest amplitude (162%) in the ionosphere above Okinawa (with high geomagnetic activity Ap of 37 nT). The most surprising finding of the study are the strong lunar tides with relative amplitudes of about 10% and the discovery of a terdiurnal lunar tide (5%) in $NmF2$ during SSW years. The periods of the ionospheric lunar tides agree with the periods of ocean tides and luni-solar variations of the atmosphere.

## 1 Introduction

It has been observed and simulated that sudden stratospheric warmings (SSWs) affect the whole atmosphere and ionosphere of the Earth (Pedatella et al., 2018). Sudden stratospheric warmings can be divided into minor and major SSWs (McInturff, 1978). Contrary to minor SSWs, major SSWs are causing a reversal of the stratospheric polar vortex from eastward to westward flow in the middle stratosphere. The central date of an SSW (SSW onset) is defined as the time point of the reversal of the stratospheric zonal wind at 10 hPa (about 30 km altitude) and at 60°N latitude. There can be also SSWs in the southern hemisphere but they are quite rare while the major SSWs in the northern hemisphere occur every 1-2 years.

In the present study, we only consider the influence of major SSWs in the northern winter hemisphere on the ionosphere. The reversal of the stratospheric polar vortex is caused by planetary wave breaking. In addition, severe disturbances, splitting or

breakdown of the stratospheric polar vortex are observed during SSW events. A major SSW is associated with a rapid increase in the polar stratospheric temperature and a reversal of the vortex from eastward to westward flow for at least 5 days (Matsuno, 1971; Schoeberl, 1978; McInturff, 1978; Matthewman et al., 2009).

It is well established that variations in polar stratospheric winds can affect mesospheric temperatures through changes in the filtering of gravity wave fluxes, which drive a residual circulation in the mesosphere. SSWs are often associated with mesospheric cooling since the breakdown of the stratospheric polar vortex leads to an enhanced gravity wave flux from below into the mesosphere with associated upwelling and adiabatic cooling of the mesosphere. In the weeks after the SSW onset, an elevated stratopause appears in the polar mesosphere. The elevated stratopause slowly descends downward during several weeks to about 50 km in altitude (Zülicke and Becker, 2013).

The present knowledge of the influences of SSWs on the ionosphere have been reviewed by Goncharenko et al. (2021). Particularly, the upward propagating solar and lunar tides could have different propagation conditions due to SSW effects on the middle atmosphere. The excitation, propagation and dissipation of solar and lunar tides have been described by means of a classical tidal theory which linearises the primitive equations providing Laplace's tidal equations (Lindzen and Chapman, 1969). The ionospheric effect of an SSW is most obvious in the low latitude ionosphere where a strongly amplified semidiurnal pattern in total electron content, equatorial vertical ion drift and equatorial electrojet have been observed (Goncharenko et al., 2021, 2013; Yamazaki et al., 2012). Yamazaki et al. (2012) also found that the geomagnetic lunar tide at the equator is enhanced in 70% of the SSW events. However, the role of lunar and solar tidal variability for generation of the ionospheric effects of SSWs is still under investigation. The relative amplitudes of the semidiurnal lunar tide ($M_2$, period 12.42 hour) is about 5% at low latitudes relative to the mean background electron density (Forbes and Zhang, 2019; Pedatella, 2014). Forbes and Zhang (2019) also mentioned that the quasi-diurnal lunar tide $O_1$ (period 25.82 hours) is important in the ionosphere. At mid-latitudes, the SSW effects on the ionosphere are more an increase of the mean electron density than ionospheric tidal variability (Mošna et al., 2021).

The present study investigates the ionospheric effects of major SSWs at the northern border of the low latitude ionosphere. We take advantage on the long-term series of ionospheric peak electron density $NmF2$ which has been observed by an ionosonde at Okinawa. Kalita et al. (2015) analyzed the seasonal and diurnal variability of $NmF2$ at Okinawa and the influence of the equatorial ionization anomaly (EIA) at this location. The attribution of the ionospheric changes to SSW events is relatively difficult since the ionospheric effect can be delayed or advanced by several days with respect to the central date of the SSW. Thus, a composite analysis of many SSW events is required for characterization of the average behaviour of the SSW-induced ionospheric changes. In addition, the solar and lunar tides are modulated by mesospheric winds. The mesospheric wind field is still not well observed but necessary for ionospheric predictions (Harvey et al., 2022). It has been observed that the migrating diurnal solar tide at the equator is reduced in the amplitude by SSW events (Siddiqui et al., 2022; Hocke, 2023). The ionosonde series allows the discussion of variability of solar and lunar tides during the SSW events. It is still an open question whether solar or lunar tidal variability is more important for the ionospheric effects of SSWs. The long-term ionosonde series recorded at several places in the world also give us a unique opportunity to study historical SSW events of the 1950s or even earlier when stratospheric observations were rare but ionospheric observations were already common.

## 2 Dataset and data analysis

The ionogram records tracings of high-frequency (HF) radio pulses reflected in the ionosphere. It is produced by the vertical incidence ionospheric sounder (ionosonde) at Okinawa (26.7°N, 128.1°E, magnetic latitude: 17.0°N), Japan. The sounding frequency is swept from 1 to 30 MHz. Ionograms are obtained at regular intervals of 15 min and contain several important factors such as the ordinary mode critical frequency or maximum reflection frequency $foF2$ of the ionospheric F$_2$ layer. In the present study, we use the manually scaled ionospheric parameter $foF2$ which is provided with a time resolution of 1 hour from 1972 to 2023 by the World Data Center for Ionosphere and Space Weather at the National Institute of Information and Communications Technology in Tokyo.

The parameter $foF2$ is closely related to the maximum electron density $NmF2$ of the ionospheric F$_2$ layer by

$$NmF2 = 1.24 \times 10^{10}(foF2)^2 \tag{1}$$

where $NmF2$ is in [m$^{-3}$] and $foF2$ in [MHz] (Davies, 1990; Chuo, 2012). In the present study, we mainly analyse the time series of $NmF2$. The relative amplitudes are computed with respect to a temporal average of $NmF2$ which is either the mean of the depicted time interval or a 60 day-lowpass filtered series of $NmF2$.

For the data analysis, we used a digital non-recursive, finite impulse response bandpass filter. Zero-phase filtering is ensured by processing the time series in forward and reverse directions. A Hamming window was selected for the filter. The number of filter coefficients corresponds to a time window of three times the central period, so that the bandpass filter has a fast response time to temporal changes in the data series. The variable choice of the filter order permits the analysis of wave trains with a resolution that matches their scale. The bandpass cut-off frequencies are at $f_c = f_p \pm 10\% f_p$, where $f_c$ is the cut off frequency and $f_p$ is the central frequency. For the semidiurnal variation, the cut-off frequencies are 1.8 and 2.2 cycles per day (cpd). Further details about the bandpass filtering are provided by Studer et al. (2012). In case of the 60 day-lowpass filter, we used the cut-off frequencies 0 and 1/(60 days).

For the case study of the SSW of 6 January 2013, we compare the Okinawa ionosonde data to data of the ionosonde at Wuhan (30.5°N, 114.4°). The Wuhan ionograms have a time resolution of 15 min, and the $foF2$ values are automatically determined.

The composite analysis is limited to major SSWs during northern hemispheric winter. The central date of a major SSW is defined as when the eastward wind changes to westward wind at 10 hPa, northward of 60°N (McInturff, 1978). The central date gives the onset time of the major SSW and are used as the timing mark for the composite analysis of major SSWs (Hocke et al., 2015). Of course, there are other processes in the mesosphere which may introduce an SSW or may follow the SSW onset. For example, an elevated stratopause can be observed in the polar region in the weeks after the SSW onset (Manney et al., 2008).

Palmeiro et al. (2023) determined the central dates of major SSWs by using the zonal wind data of the fifth-generation ECMWF atmospheric reanalysis (ERA5, Hersbach et al. (2020)). Using Table 1 of Palmeiro et al. (2023), we obtain about 31 central dates of major SSWs in the time from 1972 to 2023 (selected ionosonde series at Okinawa) fulfilling the U60 criterium

(reversal of zonal wind $U$ at 60°N). Since Table 1 of Palmeiro et al. (2023) ends in January 2021, we added the major SSW of 20 March 2022 (Vargin et al., 2022). The list of the central dates of the 29 SSW events which we use in the present study is shown in Table 1.

We also use a list of "no SSW events" with central dates from December to March in northern hemispheric winters from 1974 to 2020 when no major SSW occurred. We selected the no SSW events close to the SSW years in order to have no large difference between solar activity and geomagnetic activity of SSW years compared to no SSW years. Thus, we did not put no SSW events into the large gap of SSW events in the 1990s. The list of the 29 central dates of the "no SSW events" is shown in Table 2.

**Table 1.** Central dates of the selected 29 major SSWs in the Northern Hemisphere from 1973 to 2022. Because of some data gaps in the ionosonde series, we could not use all SSW events during this time interval which were listed by Palmeiro et al. (2023).

| | | | |
|---|---|---|---|
| 19730131 | 19770109 | 19790222 | 19800301 |
| 19810304 | 19811204 | 19840223 | 19841231 |
| 19870122 | 19871207 | 19880314 | 19981215 |
| 19990225 | 20010211 | 20011230 | 20020217 |
| 20030118 | 20040105 | 20060120 | 20070224 |
| 20080222 | 20090124 | 20100209 | 20100323 |
| 20130106 | 20180211 | 20190101 | 20210104 |
| 20220320 | | | |

**Table 2.** List of our arbitrary central dates of 29 "no SSW events" from 1974 to 2020. These dates were used for the composites of the "no SSW years". Since there were not so many no SSW years, we often put several events into one year.

| | | | |
|---|---|---|---|
| 19740313 | 19741128 | 19750313 | 19780118 |
| 19830106 | 19830131 | 19860106 | 19860301 |
| 19970211 | 19980319 | 20000124 | 20000225 |
| 20050124 | 20050211 | 20111230 | 20120101 |
| 20120217 | 20140118 | 20140302 | 20150105 |
| 20150118 | 20160104 | 20160120 | 20170101 |
| 20170118 | 20170224 | 20191204 | 20200103 |
| 20200222 | | | |

The mean geomagnetic activity Ap was 12.0 nT in the selected no SSW years and 11.7 nT in the selected SSW years. In case of solar activity, the solar radio flux at 10.7 cm (F10.7) was 122 in SSW years and smaller with 112 in no SSW years. It can be that SSW years have different conditions (e.g., solar activity, ENSO activity) than no SSW years. We performed composite

analysis for F10.7 and Ap and found no significant changes at the epoch time 0 (SSW onset time). The composite lines were flat. Thus, we are sure that the SSW-induced characteristics of the $NmF2$ composites in the present study are not influenced by solar or geomagnetic activity.

For the spectral analysis of the solar and lunar tides in $NmF2$, we calculated the fast Fourier transform (FFT) spectrum of the relative $NmF2$ variations comprising the data from 5 weeks before the central date to 5 weeks after the central date of the SSW. This data segment was folded with a Hamming window. For enhancement of the frequency resolution we performed zero padding (4 years of zeros at the begin and at the end of the data segment). The spectral analysis and the calculation of the relative amplitude were tested and calibrated by means of an artificial sine wave of a known amplitude and frequency. The length of the data interval (10 weeks) is sufficient for a separation of the solar and lunar tidal components as we have verified by means of artificial sine waves by applying the same data analysis procedure.

## 2.1 Results

We start with the case study of the major SSW of 6 January 2013. Figure 1 shows the observed $NmF2$ at Okinawa and Wuhan from 5 days before the SSW onset to 15 days after the SSW onset. Obviously, the daily cycle is quite different at both stations, though the stations just differ in latitude by 3.8 degrees. In addition, it is difficult to see a significant change in $NmF2$ which could be related to the SSW event.

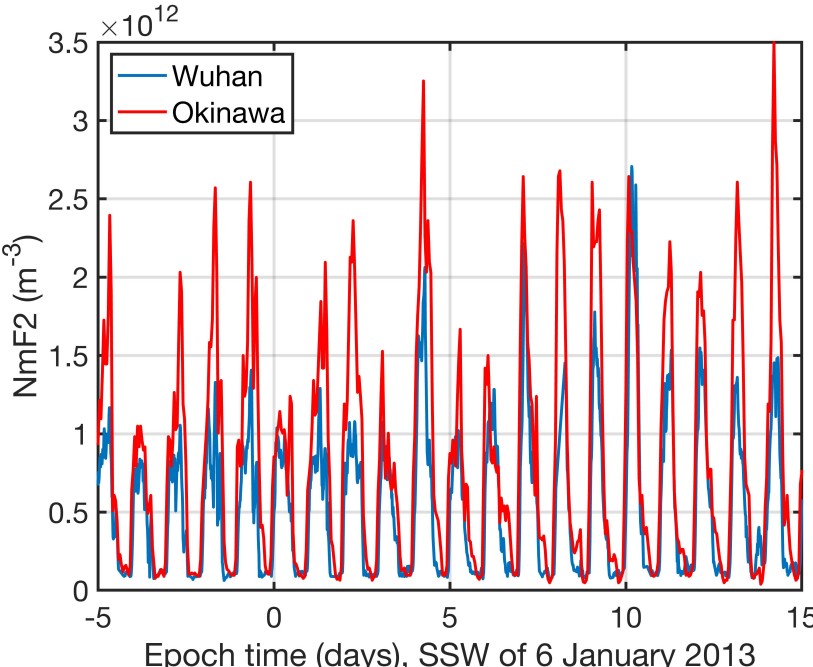

**Figure 1.** Time series of $NmF2$ at Okinawa (red) and Wuhan (blue) before and after the major SSW of 6 January 2013 (epoch time 0 corresponds to 6 January 2013).

However, Goncharenko et al. (2013) reported about a strong amplification of the semidiurnal cycle in vertical ion drift and total electron content (TEC) at the magnetic equator. This amplification of the semidiurnal pattern was maximal from 15 to 19 January 2013 which corresponds to epoch time 9 to 13 days. The maximal values of the upward ion drift were between 30 and 43 m/s (Goncharenko et al., 2013). Thus, we applied a 12 hours-bandpass to the ionosonde data at Okinawa and Wuhan. Figure 2 shows the relative amplitude of the semidiurnal cycle of $NmF2$. The relative amplitude is calculated with respect to the background mean of $NmF2$ averaged from epoch time -30 days to 50 days. The amplitude curves agree well for Okinawa and Wuhan. Both curves show a maximum at epoch time 10 days which corresponds to 16 January 2013. This maximum of the semidiurnal cycle in $NmF2$ agrees well with the date of the maximal amplification of the semidiurnal pattern in vertical ion drift and TEC at the magnetic equator reported by Goncharenko et al. (2013). Coincidently, the semidiurnal tide of the mesospheric wind at middle and high northern latitudes was enhanced as observations and simulations showed (van Caspel et al., 2023; Stober et al., 2020).

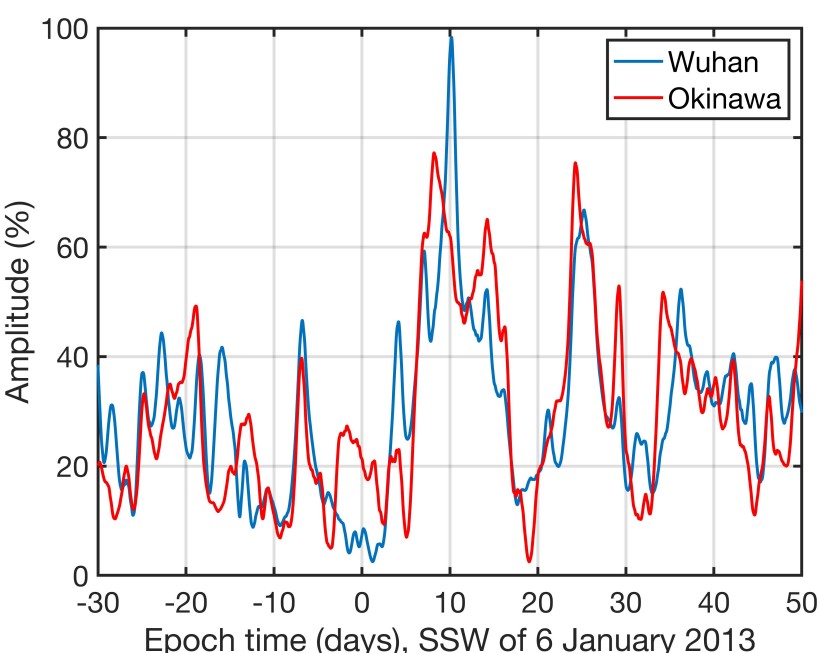

**Figure 2.** Time series of the relative amplitude of the semidiurnal cycle of $NmF2$ at Okinawa (red) and Wuhan (blue) before and after the major SSW of 6 January 2013. There is a maximum at epoch time 10 days when the largest SSW effect was observed at the magnetic equator (Goncharenko et al., 2013).

The case study of 6 January 2013 motivated us to perform a statistical study of the SSW effects in the semidiurnal cycle of the critical frequency $foF2$ from 1972 to 2023 in Okinawa. We start with $foF2$ since the composite shows a clearer effect than for $NmF2$. The difference is possibly due to the fact that the average of a series of numbers $(foF2)_i$ ($i$ is number of SSW

event) has a different behaviour than the average of the corresponding series of quadratic numbers $(foF2^2)_i$ in the composite of $NmF2$. Thus, the shapes of the composite curves are different for $foF2$ and $NmF2$.

The $foF2$ series is bandpass filtered at a central frequency of 2 cpd which correspond to the semidiurnal cycle. The amplitude series of the semidiurnal cycle is used for a composite analysis where we added 29 SSW events to a composite. Figure 3 shows the result which is averaged with a 14 day-moving average. Obviously, there is a significant peak at epoch time 0 when we calculate the composite of the SSW years (red curve). The thin lines indicate the standard error of the composite. On the other hand, the peak at epoch time 0 disappears when we compute the composite of the no SSW years (blue curve). There is also a

seasonal variation of the amplitude since the central dates of the SSWs are always in northern hemispheric winter (December to March) when the diurnal and semidiurnal variations of $foF2$ are maximal (Zolotukhina et al., 2011). The $foF2$ values are in average higher in SSW years compared to no SSW years. This is due to a higher $F2$ peak electron density in SSW years because of a higher solar activity in SSW years (F10.7=122 s.f.u. in SSW years and F10.7=112 s.f.u. in no SSW years). The geomagnetic index Ap is in average 11.7 nT in SSW years and 12.0 nT in no SSW years.

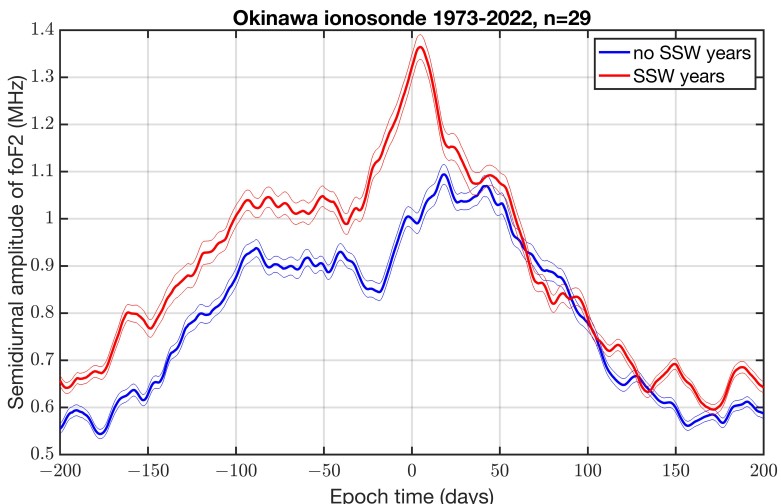

**Figure 3.** Composite of the amplitude of the semidiurnal cycle of $foF2$ at Okinawa in SSW years (red) and no SSW years (blue). The curves are averaged with a 14 days-moving average. The thin lines denote the standard errors. In total, the composite consist of 29 SSW events from 1973 to 2022.

In the following, we will analyse the relative variations of the $NmF2$ series. First, the $NmF2$ series is filtered with a 60 day lowpass in order to obtain a background series. Then, the relative variations of $NmF2$ are calculated with respect to the background series of $NmF2$. The relative variations of $NmF2$ are bandpass filtered at a central frequency of 2 cpd which correspond to the semidiurnal cycle. The relative amplitude series of the semidiurnal cycle is used for a composite analysis where we added 29 SSW events to a composite. Figure 4 shows the result which is averaged with a 14 days-moving average.

Obviously, there is a significant peak at epoch time 0 when we calculate the composite of the SSW years (red curve). On the

other hand, the peak at epoch time 0 disappears when we compute the composite of the no SSW years (blue curve). Generally, the SSW effect is not so obvious as for $foF2$. Figure 4 shows an increase of the relative amplitude from 34.0% (no SSW years) to 38.2% (SSW years). The occurrence of the peak just after epoch time 0 is a strong indication that the ionospheric effect is indeed related to the SSW onset. Please also note the sharp decrease of the amplitude by about 10% after the SSW peak at epoch time 0 was reached. We did not find such a peak when we performed the composite analysis for the diurnal variation.

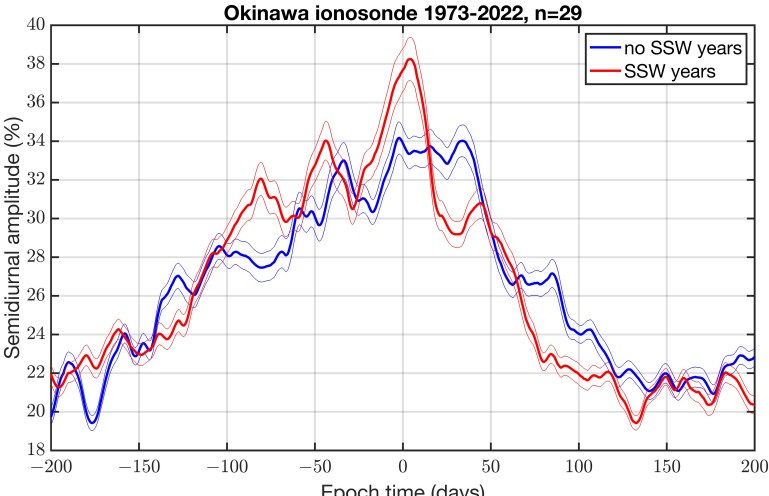

**Figure 4.** Composite of the relative amplitude of the semidiurnal cycle of $NmF2$ at Okinawa in SSW years (red) and no SSW years (blue). The curves are averaged with a 14 days-moving average. The thin lines denote the standard errors. In total, the composite consist of 29 SSW events from 1973 to 2022.

It is also interesting to study the time series of maximum relative amplitudes of the semidiurnal cycle of $NmF2$. The maximum of the relative amplitude from two weeks before the central date to two weeks after the central date of the SSW is taken. Figure 5 shows the time evolution of the maximum relative amplitude of the semidiurnal cycle since 1972 for SSW years and no SSW years. It is evident that also no SSW years can have large amplitudes in the semidiurnal cycle. In average, the relative amplitudes are 70% in no SSW years and 78% in SSW years. The largest amplitude occurred for the SSW of 31 December 1984 (about 162%). However, the geomagnetic activity was quite high (Ap=37 nT) for this event, so that the amplification of the semidiurnal cycle could be due to geomagnetic activity in this case. On the other hand, Figure 5 contains three other events of the no SSW years with high geomagnetic activity, and these three cases do not have a strong amplification of the semidiurnal amplitude. Generally, the attribution of an amplitude increase to an individual SSW event is not easy because of the high variability of the ionosphere.

We take a composite of the FFT spectrum in SSW years and no SSW years in order to get an overview about the spectral components of possible solar and lunar tides. The data segement goes from five weeks before the SSW event to 5 weeks after

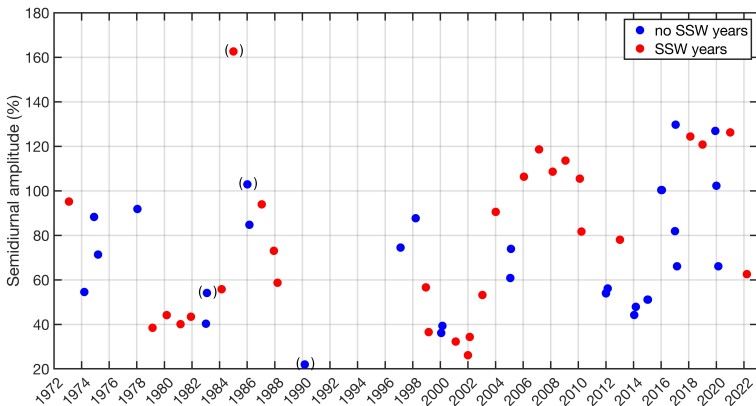

**Figure 5.** Time evolution of the maximum relative amplitude of the semidiurnal cycle of $NmF2$ at Okinawa in SSW years (red) and no SSW years (blue). The maximum is taken from two weeks before the central date to two weeks after the central date of a major SSW. The brackets around four symbols indicate a high geomagnetic activity of Kp > 4 or Ap > 27 nT.

the SSW event. Figure 6 shows the result in a zoom which does not cover the peak of the solar diurnal variation $S_1$. The $S_1$
peak is at 96.4% in no SSW years and at 95.4% in SSW years. The semidiurnal components of the sun ($S_2$) and the moon
($M_2$, period 12.42 hours) are enhanced during SSW years. In addition, we can see the quasi-diurnal lunar component ($O_1$,
period 25.82 hours) with 11.1% in SSW years and 8.8% in no SSW years. However there is a peak at the period 25 hours in
no SSW years which is close to the lunar day (24.83 hours). Possibly, for the first time, we find a lunar terdiurnal component
($MK_3$, period 8.18 hours) which is named the shallow water terdiurnal tide. The $MK_3$ component rises to 5% in SSW years.
It is evident that all lunar components in Figure 6 have periods at the periods of ocean tides (Li, 2022). Due to the interannual
variability of the tidal components, the standard error $\sigma$ of the FFT spectra is often of the same size as the enhancements during
SSW years. Figure 6 also shows a spectral peak at about 30 days which is close to the synodic period of the Moon cycle (29.53
days). At the frequency 0.5 cpd, we can see a significant peak of the two day-oscillation during no SSW years while the peak
turns into a valley during SSW years.
Finally, we like to study the time evolution of the amplification of the quasi-diurnal lunar $O_1$ tide. We derived a time series of
the $O_1$ amplitude by using a moving FFT spectra with a window length of 10 weeks, in the same manner as already described
above. Figure 7 shows an amplification of the $O_1$ tide just after the SSW onset at epoch time 0 (red curve of SSW years).

### 2.1.1   Discussion

The attribution of ionospheric effects to individual major SSWs is a difficult task at Okinawa which is in the northern low
latitude ionosphere. The case study of the SSW on 6 January 2013 suggested that the semidiurnal cycle of $NmF2$ might be
amplified after the SSW onset. Indeed, we find an increased peak of the semidiurnal amplitude on 16 January 2013 which
is coincident with the amplified semidiurnal pattern in vertical ion drift and TEC above the magnetic equator from 15 to 19

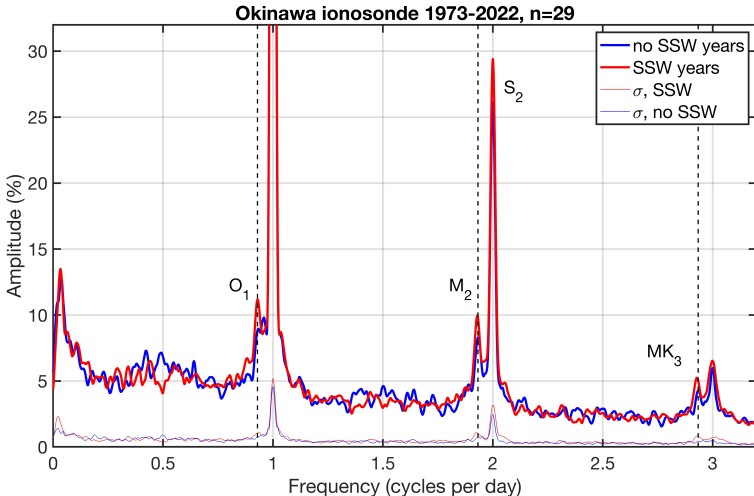

**Figure 6.** Composite of FFT spectra (time interval: five weeks before SSW to five weeks after SSW) for SSW years (red) and no SSW years (blue). The lunar components $O_1$, $M_2$ and $MK_3$ are indicated by the vertical dashed lines at the periods 25.82 hr, 12.42 hr and 8.18 hr. The solar and lunar components in the ionosphere above Okinawa are enhanced during SSW years (composite of 29 SSW events). Please see the text for information about the $S_1$ peaks. The standard errors of the spectra $\sigma$ are shown by the thin lines at the bottom.

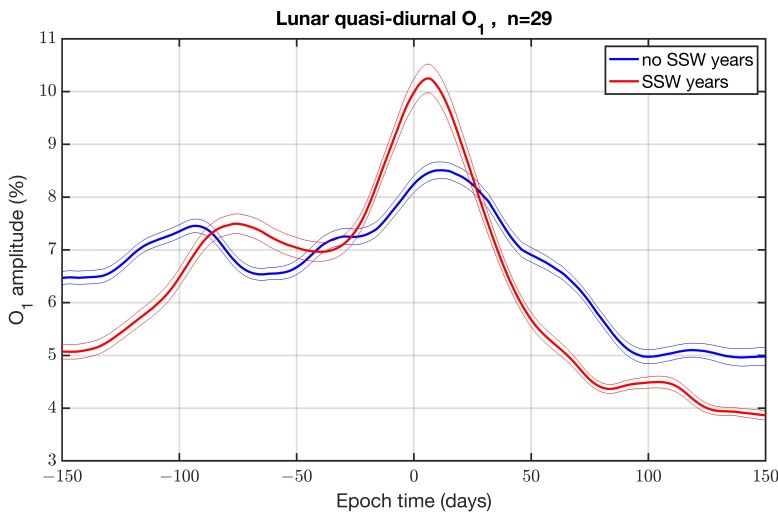

**Figure 7.** Composite of the relative amplitude of the quasi-diurnal lunar $O_1$ tide of $NmF2$ at Okinawa as function of epoch time in SSW years (red) and no SSW years (blue). The thin lines denote the standard errors. In total, the composite consist of 29 SSW events from 1973 to 2022.

January 2013 reported by Goncharenko et al. (2013). The plasma fountain in the equatorial ionosphere may slightly modulate the ionosphere above Okinawa though Okinawa is a bit northward of the equatorial ionization anomaly (EIA). The result of

190 an increased amplitude of the semidiurnal cycle of $NmF2$ at Okinawa was confirmed by a coincident amplitude increase observed by the nearby ionosonde station at Wuhan in China.

Since the ionospheric SSW effects are possibly smaller at Okinawa than at the magnetic equator, it makes sense to perform a composite analysis of 29 SSW events since 1972. The composite analysis describes the average behaviour of the ionosphere before, during and after SSW onsets. Moreover, we divided the observations into SSW years and no SSW years and calculated
composites for both cases. This idea is powerful for a critical assessment of SSW effects in spite of high ionospheric variability due to other sources such as solar and geomagnetic variability or variability of atmospheric waves from below. The composite analysis showed that there is a peak of the semidiurnal cycle of $NmF2$ which occurs at the epoch time 0 (SSW onset). This result is supported by the composite of the no SSW years which showed no peak at epoch time 0. This result is important since a peak might be due to the winter maximum of the amplitude of the semidiurnal cycle (Zolotukhina et al., 2011). Our data
analysis clearly shows that the peak cannot be explained by the seasonal variation since the SSW induced ionospheric peak amplitude does not occur in no SSW years. However, it is also a finding that the SSW effect is not so strong (relative amplitude in SSW years is 38.2% and in no SSW years 32.9%).

The long-term series of ionosonde observations are very valuable for studying the SSW effects of historical SSW events, and we found that the SSW of 31 December 1984 generated a relative amplitude of 162% in the semidiurnal cycle of the relative
$NmF2$ variations at Okinawa. In addition, we find that the amplitude increase in no SSW years also can be quite strong which should be regarded if one is analysing the ionospheric effects of individual SSW events.

The most important result of our study is contained in the composites of the FFT spectra of the 29 SSW events. The solar ($S_2$) and lunar tidal components of $NmF2$ are increased during SSW years compared to no SSW years. We find strong diurnal ($O_1$), semidiurnal ($M_2$) and terdiurnal ($MK_3$) components of the moon. The maximum of the $O_1$ amplitude occurs just after the
210 SSW onset (Figure 7). The periods of the lunar components are at the place of ocean tides (Li, 2022) and also agree well with the three periods of luni-solar variations of the atmosphere reported by Malin and Chapman (1970). In addition, the amplitudes are about 10% for $O_1$ and $M_2$ and 5% for $MK_3$ (Figure 6). This is totally different to the results which we obtain for the Kokubunji ionosonde in Tokyo (not shown here) where we do not find significant lunar components. Past studies of Forbes and Zhang (2019) and Pedatella (2014) reported about lunar diurnal and semidiurnal tides in ionospheric electron density of
215 up to 5%. In SSW years, we find the double of this value at Okinawa. An explanation might be a strong non-migrating lunar tidal components which are excited by ocean tides at the ocean-atmosphere boundary. The strong lunar tides only occur in the winter season at Okinawa. Perhaps, there exist, especially in SSW winters, special vertical propagation conditions for solar and lunar tides. The breaking or dissipation of tides in the upper mesosphere and lower thermosphere induce a tidal signal in the neutral and ion temperature at these heights which is immediately transfered by heat conduction to the ionospheric $F_2$ region.
These tidal variations in the ion temperature induce tidal variations of the ionospheric plasma and the peak electron density $NmF2$ (Zolotukhina et al., 2011).

It is interesting that the two day-oscillation in $NmF2$ is strong in no SSW years while it is missing in SSW years (Figure 6). It can be do to interactions of the tides with the quasi two day wave (QTDW) (Yue et al., 2016) or to different propagation conditions of the two day-wave from below in SSW years and no SSW years. A composite shows a peak of the QTDW in

$NmF2$ in the no SSW years in winter while the peak disappears in SSW winters (not shown in the present study). The synodic period of the Moon (29.53 days) has strong peaks in both FFT spectra. Since the data segment was relatively short (10 weeks), a longer time interval is needed for a better distinguation between the spectral peaks of the synodic period of the Moon and the 27 day-solar rotation cycle.

     Vokhmyanin et al. (2023) reported that enhanced solar activity leads to a higher occurrence rate of SSWs. Indeed, the SSW
230    years from 1973 to 2023 have an average F10.7 index of 122 s.f.u. while the no SSW years have an average value of 112 s.f.u.. This may explain that $NmF2$ at Okinawa was increased in the SSW years by 26% compared to the no SSW years (not shown in the present study). A further indication is that there were no SSWs from 1990 to 1997, and a solar minimum occurred in 1994/1995. However, the flat composites of solar or geomagnetic activity in SSW years cannot explain the peak in stratospheric temperature or the peak of the ionospheric semidiurnal amplitude (Figure 4) at the SSW onset time. On the other hand, it is
interesting to better understand why the mean state of the ionosphere is different in SSW years compared to no SSW years, correlated changes in ENSO and QBO may also play a role (Vokhmyanin et al., 2023).

## 3    Conclusions

We find a clear SSW effect in the $foF2$ series at Okinawa (Figure 3). In the composite of 29 SSW events, the amplitude of the semidiurnal cycle of $foF2$ shows a peak at the SSW onset time in SSW years. This peak is certainly not due to seasonal
effects since the peak disappears in the composite of no SSW years. In so far, our analysis method is powerful for a critical assessment and attribution of SSW effects in the highly variable ionosphere. Composites of solar and geomagnetic activity showed no relevant variations at the SSW onset time, but the solar activity was generally higher in the SSW years than in the no SSW years. The composite of the relative $NmF2$ variations also shows a significant peak of the semidiurnal amplitude at the SSW onset time in the SSW years (Figure 4).

The evolution of the amplification of the semidiurnal amplitude of the relative $NmF2$ variations in winter is shown from 1972 to 2023 (Figure 5). It is evident that the amplification of the semidiurnal amplitude also can occur in no SSW years. Thus, the major SSWs cannot be the sole reason for a sudden amplification of the semidiurnal amplitude in the ionosphere above Okinawa. Other processes such as minor sudden stratospheric warmings or variability of the mesospheric polar vortex may play a role in the years without major SSWs.

The solar and lunar tidal components at Okinawa are slightly enhanced during SSW years compared to no SSW years (Figure 6). For the first time, we find a lunar terdiurnal component with a relative amplitude of about 5% in SSW years. The lunar diurnal and semidiurnal components have relative amplitudes of about 10% in SSW years. The periods of the lunar ionospheric tidal variations are at the periods of ocean tides and luni-solar variations of the atmosphere (Li, 2022; Malin and Chapman, 1970). Our study showed that the long-term series of $foF2$ and $NmF2$ from ionosondes are valuable for learning
about ionospheric effects of historical SSWs and the study of lunar tides which are excited in the ocean-atmosphere system.

*Code availability.* Matlab codes can be provided upon request.

*Data availability.* The Okinawa ionosonde data are provided by the WDC for Ionosphere and Space Weather, Tokyo, National Institute of Information and Communications Technology at https://wdc.nict.go.jp/IONO/HP2009/ISDJ/manual_txt-E.html. The Wuhan ionosonde data are provided by the meridian project at https://data.meridianproject.ac.cn. Data about solar and geomagnetic activity were provided by NASA's Omniweb at https://omniweb.gsfc.nasa.gov/.

*Author contributions.* Concept of the study: all; data analysis: G.M and K.H.; writing: K.H., corrections and discussion of the manuscript: all.

*Competing interests.* No competing interests.

*Acknowledgements.* We thank the WDC for Ionosphere and Space Weather, Tokyo, National Institute of Information and Communications Technology for providing the long-term $foF2$ series at Okinawa and for operation of the ionosonde. We also thank the Meridian project for providing the ionosonde data at Wuhan. Open access funding was provided by the University of Bern and swissuniversities. The reviewers are thanked for valuable comments and improvements.

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
