# Peer review of "Influences of sudden stratospheric warmings on the ionosphere above Okinawa"

_EGUsphere, 2024_

## Author Response (AR1)

**Point-to-point response to Reviewer 1:**

Thank you very much for your careful review, critical comments and improvements! We performed a moderate revision. The point-to-point response is quite similar to our previous reply in the discussion forum. Table 1 and Table 2 contain all central dates which we used in the composites of SSW and no SSW years. Figures 3 to 6 are slightly modified since we removed a bug in the "no SSW" list (Table 2). In the revised manuscript, we added Figure 7 which shows that the lunar O1 tide is maximal, shortly after the SSW onset.

**Your points are in bold face**, *our response is in italic font style*, the changed text is in a normal font style.

**1) In order to understand variability that is not related to SSWs, the authors perform a composite analysis of the variability in years without SSWs. Though I agree with the approach that the authors have taken, I believe that they should provide additional details about the no SSW events. The no SSW years are referenced to a central date (e.g., Figures 3 and 4); however, in the absence of a SSW it is not clear how the central date is determined. The authors should provide additional details on how the central date is defined for the no SSW cases. Additionally, for scientific reproducibility, I suggest that the authors include a listing of the years that are used for the no SSW cases.**

*We agree and we will provide tables of the central dates for the selected 29 SSW events and the 29 no SSW events in the revised manuscript. The central dates of the no SSW events are selected arbitrarily between December and March. Often we took month and day of a true SSW event and shifted it to a no SSW year. We tried to keep the distribution of the no SSW events similar to those of the SSW events. Thanks to your comment, we found a small error in the no SSW list since we used one date two times. As a consequence , we removed this shortcoming and replotted the figures new. The changes are small in the new version.*

Table 1 and Table 2 of the revised manuscript contain all SSW central dates of the SSW years and the no SSW years.
Figures 3 to 6 are reprocessed without the bug in the previous list of no SSW events.

**2) In lines 123-124, the authors state that "we start with foF2 since the composite shows a clearer effect than for NmF2". Since foF2 and NmF2 are directly related to each other, one would expect that the results for foF2 and NmF2 are also similar, though perhaps scaled differently since $NmF2 \sim foF2^2$. Furthermore, it is unclear to the reviewer why having foF2 being the original measurement (lines**

124-125) would impact the results since the conversion from foF2 to NmF2 does not introduce additional errors. The authors should provide additional explanation as to why the conversion from foF2 to NmF2 influences the results.

*The different shapes of the composite curves of foF2 and NmF2 have been a puzzle for us, and we controlled our programs with a lot of effort. In so far, we are sure that this result is robust and true. In the revised manuscript, we explain the different composite curves by the following argument:*

We start with $foF2$ since the composite shows a clearer effect than for $NmF2$. The difference is possibly due to the fact that the average of a series of numbers $(foF2)_i$ ($i$ is number of SSW event) has a different behaviour than the average of the corresponding series of quadratic numbers $(foF2^2)_i$ in the composite of $NmF2$. Thus, the shapes of the composite curves are different for $foF2$ and $NmF2$.

**3) The authors state that "the SSW effect [in NmF2] is not so obvious as for foF2" (line 142). However, Figure 3 shows the absolute values of foF2 and Figure 4 shows the relative values for NmF2. It is therefore difficult to directly compare the results. Would the enhancements be similar if both are shown in either absolute or percentage variations?**

*Please find in the discussion forum the composite of the absolute NmF2 values. The peak in the NmF2 composite is not so strong as in the composite of foF2. We added an explanation for the different behaviour of the composites of the foF2 and NmF2 series*

The difference is possibly due to the fact that the average of a series of numbers $(foF2)_i$ ($i$ is number of SSW event) has a different behaviour than the average of the corresponding series of quadratic numbers $(foF2^2)_i$ in the composite of $NmF2$. Thus, the shapes of the composite curves are different for $foF2$ and $NmF2$.

**4) In determining the results for Figure 5, do the authors consider the effects of geomagnetic activity? Although such effects may be small in the composite analysis, for any individual event it is possible that geomagnetic effects could large. It is recommended that the authors consider only using geomagnetically quiet days when determining the maximum amplitude in any given year.**

*We agree. In the revised manuscript we indicate the SSW events with high geomagnetic activity (Kp > 4). There is only one true SSW event with high Kp value (31.12.1984). Interestingly, this event showed the maximal amplification of the semidiurnal amplitude of NmF2. There are three events in the no SSW*

*list but these cases showed low or moderate amplification of the semidiurnal amplitude of NmF2.*

Figure 5 is modified and shows brackets around the symbols of the events with high geomagnetic activity.

**Point-to-point response to Reviewer 2:**

Thank you very much for your careful review, critical comments and improvements! We performed a moderate revision. The point-to-point response is quite similar to our previous reply in the discussion forum. Table 1 and Table 2 contain all central dates which we used in the composites of SSW and no SSW years. Figures 3 to 6 are slightly modified since we removed a bug in the "no SSW" list (Table 2). In the revised manuscript, we added Figure 7 which shows that the lunar O1 tide is maximal, shortly after the SSW onset.

**Your points are in bold face**, *our response is in italic font style*, the changed text is in a normal font style.

**1) Line 18: Please be more specific with regards to the reversal of the zonal mean zonal wind of the stratospheric polar vortex.**

*We added a more detailed description:*

Contrary to minor SSWs, major SSWs are causing a reversal of the stratospheric polar vortex from eastward to westward flow in the middle stratosphere.

**2) Line 141: The authors mention that the effect of SSW is not so obvious in NmF2 as for fof2. It will be better if the authors could elaborate on the reason behind this statement as it is not discussed anywhere later in the manuscript.**

*The different shapes of the composite curves of foF2 and NmF2 have been a puzzle for us, and we controlled our programs with a lot of effort. In so far, we are sure that this result is robust and true. In the revised manuscript, we explain the different composite curves by the following argument:*

We start with $foF2$ since the composite shows a clearer effect than for $NmF2$. The difference is possibly due to the fact that the average of a series of numbers $(foF2)_i$ ($i$ is number of SSW event) has a different behaviour than the average of the corresponding series of quadratic numbers $(foF2^2)_i$ in the composite of $NmF2$. Thus, the shapes of the composite curves are different for $foF2$ and $NmF2$.

Minor corrections:
Line 43: latitudes - corrected now

Line 56: whether - corrected now

Line 82: bracket is missing in the citation of McInturff (1978) - corrected now